# Foliar Applications of Humic Substances Together with Fe/Nano Fe to Increase the Iron Content and Growth Parameters of Spinach (*Spinacia oleracea* L.)

Metin Turan [1], Melek Ekinci [2], Raziye Kul [2], Ayhan Kocaman [3], Sanem Argin [1], Anastasia M. Zhirkova [4], Irina V. Perminova [4] and Ertan Yildirim [2,*]

[1] Department of Agricultural Trade and Management, Faculty of Economy and Administrative Sciences, Yeditepe University, 34250 Istanbul, Turkey

[2] Department of Horticulture, Faculty of Agriculture, Atatürk University, 25240 Erzurum, Turkey

[3] Environmental Engineering Department, Engineering Faculty, Karabük University, 78050 Karabuk, Turkey

[4] Department of Chemistry, Lomonosov Moscow State University, 119991 Moscow, Russia

* Correspondence: ertanyil@atauni.edu.tr

**Abstract:** Iron deficiency, which severely decreases the plant yield and quality, is one of the major problems of calcareous soils. Foliar applications of humic substances and/or Fe fertilizers are environmentally friendly methods to cope with Fe deficiency. The aim of this study was to investigate the combined effect of Fe/nano Fe and humic/fulvic acid-based biostimulant foliar applications on the Fe content and plant growth parameters of spinach. Treatment solutions were prepared either by mixing a common Fe fertilizer, $FeSO_4 \cdot 7H_2O$, with different commercial biostimulants (Fulvic-based: Fulvagra®, Fulvagra®WSG; Humic-based: HS300®, Humin Fe® and Liqhumus®, Grevenbroich, Germany) or by mixing nano ferrihydrite with different ratios of fulvic substance (FA-50, FA-75, and FA-100) and humic acid (Nano Iron). Growth parameters (plant fresh and dry weights, plant dry matter, root fresh and dry weights, root dry matter, leaf number per plant, and leaf area); chlorophyll reading value (SPAD); chlorophyll (a,b, and total) and carotenoid contents; and leaf and root mineral contents (N, P, K, Ca, Mg, S, Cu, Mn, Zn, B, active Fe, and total Fe) of samples were determined. Our results showed that foliar application of biostimulants together with Fe sources improved the nutrient uptake, chlorophyll contents, growth characteristics, and yield; however, not all humic substances had the same effect. When all parameters were considered, Fulvagra treatment—which contained 17% fulvic acid and microorganisms in its content together with 20 mM $FeSO_4 \cdot 7H_2O$—was the most effective application, followed by FA100 treatment containing fulvic acid and 20 mM nano ferrihydrite. This finding indicates that fulvic acid containing biostimulants is more effective in foliar applications than humic-based biostimulants against Fe deficiency due to their low molecular weight which enables better penetration into the leaves. In conclusion, foliar applications of fulvic substances together with Fe fertilizers can be used to increase the Fe uptake of crops and the yields under Fe-deficient conditions.

**Keywords:** iron; fertilization; spinach; biostimulants; humic acid; fulvic acid; nano Fe; Fe chelate; active Fe; total Fe

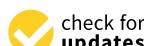



## 1. Introduction

Plant nutrients play an essential role in plant metabolism and the balance among them positively stimulates the yield and quality of plants [1]. Excess or deficient uptake of micro- and macroelements causes a number of metabolic disorders in the plant, which ultimately results in a lower yield. Iron is an important microelement because it forms iron–sulphur clusters within ferredoxin, a protein found in the chloroplasts which functions to mediate electron transfer during photosynthesis. Thus, deficiency of Fe induces

abnormalities in chloroplast morphology/structure and causes reduced chlorophyll contents/photosynthetic rate as well as diminished respiratory ability, all of which severely decrease the plant yield and quality [2–7]. However, growth reduction in Fe-deficient plants is not the only result of a decrease in chlorophyll synthesis, electron transport, and $CO_2$ assimilation. Other aspects of plant metabolism are also affected due to the contribution of Fe to the structure and activity of various enzymes [1,8]. Fe deficiency is one of the major problems of calcareous soils, which accounts for one-third of the cultivated areas in the world [9]. Although Fe is the fourth most abundant element in the lithosphere (5% of the earth's crust), the amount of plant-available Fe in the soil may vary depending on the soil material and environmental factors, such as soil pH, the redox potential of soil, the presence of carbonates/bicarbonates in the environment, and high concentrations of phosphorous. Iron deficiency is usually observed in alkaline or over-limed soils [5] because at high pH, Fe is oxidized into insoluble ferric ($Fe^{3+}$) forms, predominately ferric oxides. The amount of soluble Fe reaches the minimum level at pH 6.5–8.0. At lower pH, Fe becomes more available for plant uptake. Iron is absorbed by plant roots in the form of $Fe^{2+}$ (ferrous) ions, which are transported to the root surfaces via Fe chelates [10,11]. The presence of free $Mn^{2+}$, $Cu^{2+}$, $Zn^{2+}$, $Ca^{2+}$, $Mg^{2+}$, or $K^+$ ions in the soil might also negatively affect the Fe uptake. Especially in the environments where $Mn^{2+}$ and $Cu^{2+}$ ions are concentrated, Fe deficiency occurs due to the antagonistic effects of these ions [10]. Additionally, the presence of pollutants, such as $Cd^{2+}$ and $Co^{2+}$, in the soil has also been shown to interfere with Fe uptake by preventing the transport of Fe from roots to shoots [12].

Humic substances are final products that are formed due to decomposition of dead biomass in the soil, and these substances are the most natural substances on earth. The main tasks of humic substances (humic, fulvic, and himetomelanic acid, etc.) include controlling nutrient status of the soil, overseeing the exchange of carbon and oxygen between soil and atmosphere, converting toxic chemicals by breaking them down, and finally, preparing the necessary media for the survival of soil organisms. Humic substances are also very effective on plant physiology. Humic substances are composed of humic acids that can be dissolved only in an alkaline environment, fulvic acids that dissolve in both acid and alkaline environments, and humins that cannot be dissolved. The most obvious difference between humic and fulvic acids is that the molecular size of humic acids is much larger compared with fulvic acids. In fact, humic acids can be 100 thousand times larger than fulvic acids. The humic and fulvic acid content of the products varies according to the sources from which they are obtained. The products obtained from leonardite, an immature type of lignite, are usually soluble in alkaline environments and contain high levels of humic acid and low levels of fulvic acid. Products of herbal lignin origin are acidic and contain high levels of fulvic acid and low levels of humic acid.

Due to the fact that almost 30% of the arable soil in the world is calcareous and alkaline [13], the availability of Fe becomes an important issue for the food chain because it determines the plant yield/productivity and Fe concentration of edible tissues, both of which affect human nutrition. Consuming Fe-deficient plants may result in Fe deficiency in humans [14], which causes serious health problems. It is possible to remedy Fe deficiency by applying inorganic Fe sources and Fe chelates to the soil or leaves [7,13,15,16]. However, the high prices of Fe fertilizers and high solubility of synthetic Fe chelates (which, under field conditions, cause leaching from rhizosphere) make them uneconomical and environmentally unfriendly [17,18].

Foliar applications are a greener alternative to soil amendments to cope with Fe deficiency. An emerging method is to use nano Fe complexes in soil treatments and foliar applications to increase the crop yield and prevent Fe deficiency [2,19–24]. Application of micronutrient nano fertilizers, such as FeO and $Fe_2O_3$, had promising results in terms of leaf chlorophyll content and root dry mass [25]. A study on the agricultural use of nanoparticles reported that magnetite ($Fe_3O_4$), ferrihydrite ($FeOOH \bullet \times H_2O$), and hematite ($-Fe_2O_3$) treatments considerably enhanced the total N of roots or shoots, number of pods, dry weight of pods, number of seeds, and bean yield in common beans [26]. Although more



studies are needed, nano-sized fertilizers might improve the nutrient use efficiency and reduce fertilizer-dosage-related environmental problems. Another approach to cope with Fe deficiency is to apply humic substances (HS). In many studies, when applied to the soil or to the leaves, environmentally friendly humic substances have been shown to increase the Fe uptake and the plant growth by acting as chelators and biostimulants [27–39]. Although HS and/or Fe fertilizers/chelates were shown to improve Fe uptake via leaf sprays, the number of studies on foliar applications of nano Fe together with HS is limited [40]. The aim of this study was to investigate the combined effect of Fe/nano Fe and humic/fulvic acid-based biostimulant foliar applications on the Fe content and plant growth parameters of spinach (*Spinacia oleracea* L.) in Fe-deficient soil. Both seed pre-treatments and foliar applications were performed on all samples. Spinach was chosen as the plant material because it is a nutritious green leafy vegetable with high Fe content and is widely consumed around the world.

## 2. Materials and Methods

### 2.1. Treatments

The pot experiments were carried out in the greenhouses of The Plant Production, Application and Research Centre of Atatürk University, Erzurum, Turkey, in October 2020. *Spinacia oleracea* L. cv. Matador, which is widely grown in Turkey, was used as plant material. Commercial biostimulants (Fulvic acid based: Fulvagra®, Fulvagra®WSG; Humic acid based: HS300®, Humin Fe®, Liqhumus® (LH), Grevenbroich, Germany) were provided from Humintech GmbH. Fulvic acid-based samples containing nano ferrihydrite and different ratios of fulvic acid ligands (FA-50, FA-75 and FA-100) and humic acid-based sample containing nano ferrihydrite (Nano Iron) were synthesized by Prof. Perminova's Laboratory in the Department of Chemistry at the Lomonosov Moscow State University (LMSU), Russia. The content and physical properties of commercial biostimulant products and nano Fe samples with particle size of 2–5 nm are given in Table 1. Four doses of Fe (0, 5, 10, and 20 mM) were applied to determine the most appropriate Fe dose for spinach.

**Table 1.** Treatments, contents of biostimulants, and iron products applied.

| Treatment | Biostimulant | Iron Type | Content and Physical Properties of Biostimulants and Nano Iron |
|:---:|:---:|:---:|:---:|
| 1 | Fulvagra® (Humintech GmbH) * (Fulvic acid-based) | $FeSO_4 \cdot 7H_2O$ | Content % dry wt: 23–25, Fulvic acid: 17%, Humic acid: 1.0%, Organic substance: 20–21% Microorganisms ($1 \times 10^9$ cfu·mL$^{-1}$): *Azospirillum brasilence* spp., *Bacillus pumulis*, *Bacillus megaterium*, *Bacillus subtilis* pH: 8–9 |
| 2 | Fulvagra®WSG (Humintech GmbH) * (Fulvic acid-based) | $FeSO_4 \cdot 7H_2O$ | Content (% dry wt): 90–95, Fulvic acid: 63–64%, Humic acid: 9–10.0%, Organic substance: 75–80% pH: 8–9 |
| 3 | HS300® (Humintech GmbH) * (Humic acid-based) | $FeSO_4 \cdot 7H_2O$ | Content (% dry wt): 27–30, Humic acid: 21–22%, Fulvic acid: 5–6%, Organic substance: 28% Particle size of insoluble constituents: <5 μm pH: 4–5 |
| 4 | Humin Fe® (Humintech GmbH) * (Humic acid-based) | $FeSO_4 \cdot 7H_2O$ | Content (% dry wt): Potassium humates: 80–85%, Potassium as $K_2O$: 10–12%, Total organic nitrogen: 1.0%, Others: 2.0% Particle size of insoluble constituents: <100 μm pH: 9–10 |

**Table 1.** *Cont.*

| Treatment | Biostimulant | Iron Type | Content and Physical Properties of Biostimulants and Nano Iron |
|---|---|---|---|
| 5 | Liqhumus® (Humintech GmbH) * (Humic acid-based) | $FeSO_4 \cdot 7H_2O$ | Content (% dry wt): 22–23, Humic substance: 18–20%, Potassium as $K_2O$: 2.5–3.0%, Total organic nitrogen: 1.0% Particle size of insoluble constituents: <100 μm pH: 9–10 |
| 6 | FA-50 (LMSU) ** (Fulvic acid-based) | Nanoferrihydrite ** | Content (% dry wt): 88–89, Fulvic acid: 33.0–33.5%, Maltodextrin: 33.0–33.5% Organic substance: 66.0–67.0%, Ash content: 5.0%, Iron: 17.0% pH: 9–10 |
| 7 | FA-75 (LMSU) ** (Fulvic acid-based) | Nanoferrihydrite ** | Content (% dry wt): 88–89, Fulvic acid: 50%, Maltodextrin: 16.6–17.6% Organic substance: 66.6–67.6%, Ash content: 5.0%, pH: 9–10 |
| 8 | FA-100 (LMSU) ** (Fulvic acid-based) | Nanoferrihydrite ** | Content (% dry wt): 88–90, Fulvic acid: 67.5–69.5%, Organic substance: 67.5–69.5%, Ash content: 5.0%, pH: 9–10 |
| 9 | Nano Iron (LMSU) ** (Humic acid-based) | Nanoferrihydrite ** | Content (% dry wt): 88–90, Humic acid: 73–75%, Organic substance: 73–75%, Ash content: 5.0%, pH: 9–10 |
| Control | – | – | |

* Humintech GMBH, Am Pösenberg 9–13, Grevenbroich/Germany; ** Lomonosov Moscow State University (synthesized as described in Zimbovskaya et al., 2020).

## 2.2. Characterization of the Nano Ferrihydrite Samples Used in This Study

The fulvic acid-based samples containing nano ferrihydrite were characterized with X-ray diffraction (D-MAX 2500, Rigaku, Japan), TEM (Libra 200 MC microscope, Zeiss, Oberkochen, Switzerland), Mössbauer spectroscopy (MS1104EM Express Mössbauer spectrometer, Cordon GmbH, Rostov-on-Don, Russia). The X-ray diffraction patterns of the obtained fulvate and fulvate–polymaltose complexes of iron (III) hydroxide are shown on Figure 1a. Analysis of the diffraction patterns showed that all samples are characterized by low crystallinity of the Fe-containing phase, which manifests itself in the absence of resolved high-intensity peaks. In all diffraction patterns, an intense peak at 2θ 36 can be noted, which can be attributed to the ferrihydrite phase based on data from the PDF-2 database. Peaks at 2θ 31.7 and 45.5 can be attributed to the presence of a small amount of the NaCl phase. The obtained TEM images of iron fulvate FA-100 and iron fulvate–polymaltose FA-50 are shown in Figure 1b,c, respectively. According to the TEM data for preparations FA-100, FA-75, and FA-50, discrete nanoparticles 2–5 nanometers in size are observed. The forms of existence of Fe in the composition of the obtained complexes were additionally studied by the method of Mössbauer spectroscopy (Figure 1d). Survey spectra were obtained for all samples in a wide range of velocities at room temperature, from which it follows that the samples predominantly contain a paramagnetic doublet related to Fe (III) compounds in an octahedral oxygen environment.

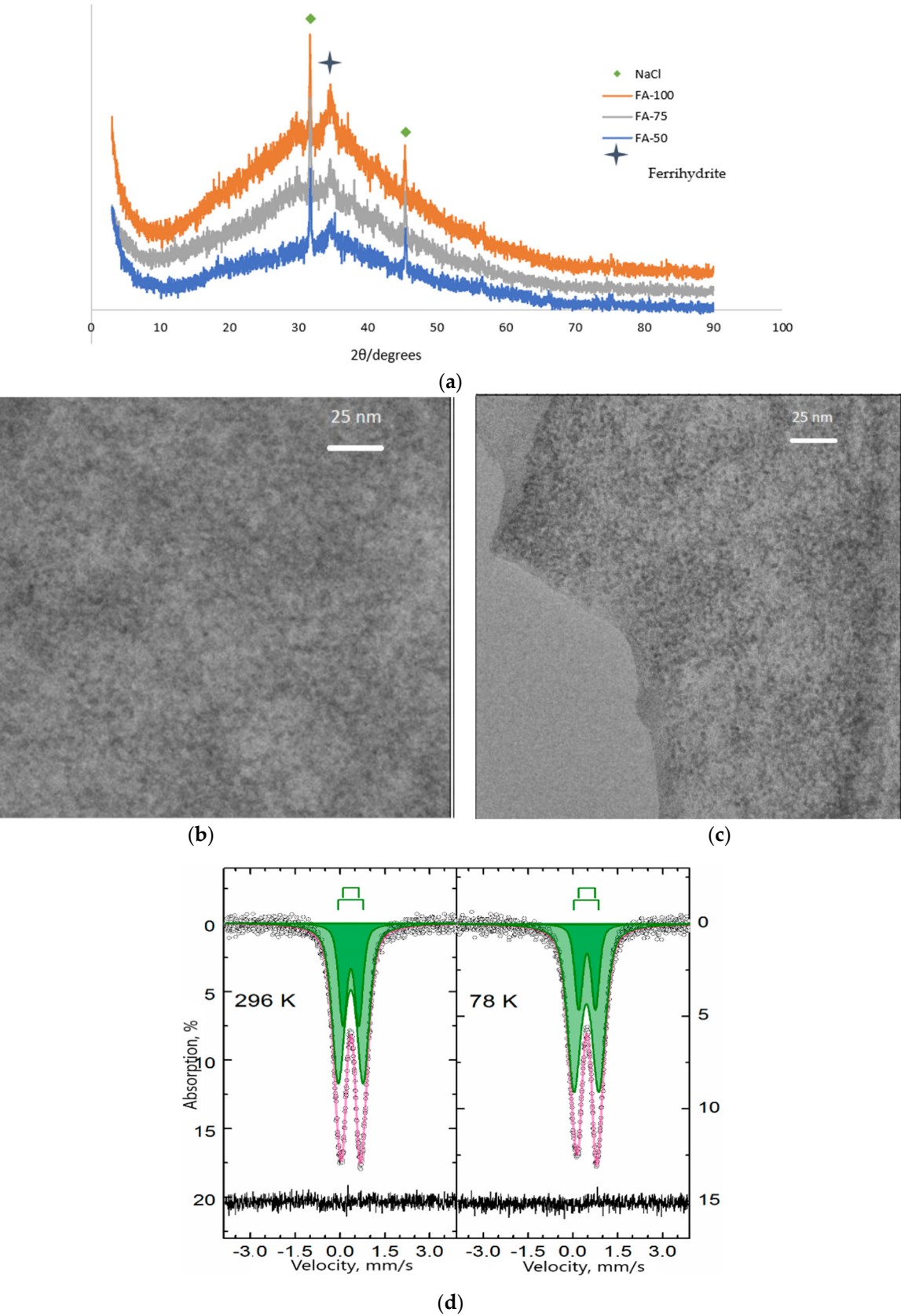

**Figure 1.** Characterization of Nano Iron samples (**a**) X-ray diffraction patterns: FA-100 is highlighted with orange; FA-75—gray; FA-50—blue colors. The peaks of NaCl are marked with green rhombuses. The ferrihydrite broad peak is marked with an asterisk, (**b**) TEM image of FeOOH in fulvic acid (FA-100), (**c**) TEM image of FeOOH in fulvic acid and maltodextrin (FA-50), (**d**) The Mössbauer spectra of fulvic acid FeOOH at room and liquid nitrogen temperatures.

### 2.3. Experiment Setup

The experiment was carried out according to a completely randomized factorial design with 4 doses (0, 5, 10, and 20 mM Fe), 3 replications (3 pots in each replication), and 10 treatments for a total of 360 pots. Pots (1.5 L), filled with a mixture of soil and sand (2:1, *v/v*), were labelled and placed on benches in the greenhouse. Treatment solutions were prepared either by mixing a common Fe fertilizer, $FeSO_4 \cdot 7H_2O$, with different commercial biostimulants (Fulvagra®, Fulvagra®WSG, HS300®, Humin Fe®, Liqhumus®, Grevenbroich, Germany) or by mixing nano ferrihydrite with fulvic or humic substances, as described in Table 1. In order to determine the most efficient Fe dose, biostimulants were first homogenized with 5, 10, and 20 mM Fe fertilizer (either $FeSO_4 \cdot 7H_2O$ or nano ferrihydrite). In this study, humic acid and fulvic acid were used as carriers of iron in the removal of Fe deficiency by foliar applications. In nano iron applications, humic and fulvic acids are used as carriers, whereas the iron source used as $FeSO_4 \cdot 7H_2O$ is used by bringing it to nano dimensions. Spinach seeds were kept in fertilizer solutions prepared at different Fe doses for 2 h. The seeds of the control group were kept in distilled water for the same amount of time. Pre-treated seeds were sowed as 5 seeds per pot at a depth of 2 cm. The soil properties are given in Table 2. Irrigation was accomplished with tap water right after sowing, and care was taken to keep the soil moist at a level close to the field capacity during the experiments. Seed pre-treatment solutions were also used for foliar applications. First, foliar application was performed approximately two weeks after sowing and repeated 4 times during the experiment period (65 days). Plants were kept under natural daylight (approximate daily temperatures of 25 °C and 50% relative humidity). After the efficient dose was determined based on regression analysis, the Fe fertilizer concentration in the treatments was fixed to 20 mM.

**Table 2.** Soil properties.

| pH | EC ($\mu$S cm$^{-1}$) | OM | N (%) | P (mg kg$^{-1}$) | K (mg kg$^{-1}$) | Ca (mg kg$^{-1}$) |
|---|---|---|---|---|---|---|
| 7.33 | 42.00 | 0.33 | 0.001 | 3.67 | 245.33 | 1552.67 |
| Mg (mg kg$^{-1}$) | Zn (mg kg$^{-1}$) | Fe (mg kg$^{-1}$) | Mn (mg kg$^{-1}$) | Cu (mg kg$^{-1}$) | B (mg kg$^{-1}$) | Na (mg kg$^{-1}$) |
| 44.33 | 0.33 | 1.67 | 1.45 | 1.22 | 0.20 | 10.40 |

### 2.4. Sample Preparation

Plants that reached harvest size approximately 65 days after sowing were separated into shoots (plant) and roots for further measurements. Harvested leaves and roots were immediately weighed (for the fresh leaf and root weights) and recorded. Leaf areas were measured with a portable leaf area meter (CI-202, Laser Leaf Area Meter, CID), and the per-plant average value was calculated. Samples were dried in a forced-air oven at 68 °C for 72 h to obtain leaf and root dry weights. Dried leaf and root samples were ground for mineral element analysis.

### 2.5. SPAD, Chlorophyll a, Chlorophyll b, Total Chlorophyll, and Carotenoid Analysis

Leaf chlorophyll contents were measured from fully opened middle leaves with a hand-held chlorophyll meter (SPAD-502, Konica Minolta Sensing, Inc., Osaka, Japan), and the average of 6 replicates was recorded as the SPAD value [41].

To determine the leaf chlorophyll and carotenoid concentration of plants, leaf discs with a diameter of 10 mm were obtained from the leaf sample using a punch. The leaves were placed in 2 mL Eppendorf tubes with an iron ball inside and crushed with 0.2 mL cold acetone (80%) in a tissue shredder (TissueLyser LT, Qiagen, Hilden, Germany) at 50 Hz for three minutes. It was then brought to the final volume (2 mL) with 80% cold acetone and centrifuged at 10,000 rpm at 5 °C. Finally, absorbance values at 663, 645, and 450 nm were measured using a microplate spectrophotometer (Thermo Scientific™

Multiskan™ FC Microplate Photometer, Vantaa, Finland). To determine the leaf chlorophyll and carotenoid concentration of plants, leaf discs with a diameter of 10 mm were obtained from the leaf sample using a punch. The leaves were placed in 2 mL Eppendorf tubes and an iron ball was placed inside and crushed with 0.2 mL 80% cold acetone in a tissue shredder at 50 Hz for three minutes (TissueLyser LT, Qiagen, Hilden, Germany). It was then brought to the final volume (2 mL) with 80% cold acetone and centrifuged at 10,000 rpm at 5 °C. Finally, absorbance values at 663, 645, and 450 nm were measured with a microplate spectrophotometer (Thermo Scientific™ Multiskan™ FC Microplate Photometer). Then, the chlorophyll a, chlorophyll b, total chlorophyll, and carotenoid contents were calculated as mg cm$^{-2}$ according to the method of Arnon [42] and Wellburn [43].

$$\text{Chla (mg·cm}^{-2}) = [(12.7 \times A_{663}) - (2.6 \times A_{645})] \times (\text{mL of Acetone 80\%})/(\text{Leaf Area (cm}^2)) \times 10$$

$$\text{Chlb (mg·cm}^{-2}) = [(22.9 \times A_{645}) - (4.68 \times A_{663})] \times (\text{mL of Acetone 80\%})/(\text{Leaf Area (cm}^2)) \times 10$$

$$\text{Chlt (mg·cm}^{-2}) = [(20.2 \times A_{645}) + (8.02 \times A_{663})] \times (\text{mL of Acetone 80\%})/(\text{LeafArea (cm}^2)) \times 10$$

$$\text{Ct (mg·cm}^{-2}) = [(1000 \times A_{450}) - (1.9 \times \text{Chla}) - (63.14 \times \text{Chlb})]/214 \times 10$$

### 2.6. Soil Analysis

Soil samples were air dried, crushed, and passed through a 2 mm sieve before physical and chemical analyses were conducted. The Kjeldahl method [44] was used to determine organic N, whereas plant-available P was determined by using the sodium bicarbonate method of Olsen et al. [45]. Electrical conductivity (EC) was measured in saturation extracts according to Rhoades [46]. Calcium carbonate concentrations were determined according to McLean [47] and soil pH was measured in 1:2 extracts. Soil organic matter was determined using the Smith–Weldon method according to Nelson and Sommers [48]. Ammonium acetate buffered at pH 7 [49] was used to determine exchangeable cations. Microelements in the soils were determined by diethylenetri–aminepentaacetic acid (DTPA) extraction methods [50].

### 2.7. Plant Leaf and Root Nutrient Content Analyses

For the determination of the nutrient concentrations in leaves and roots, 200 mg ground samples were digested using a mixture of 3 mL of nitric acid ($HNO_3$) (65%, Emplura, Merck, Darmstadt, Germany) and 3 mL of hydrogen peroxide ($H_2O_2$) (35%, Emprove, Merck, Germany) in a microwave oven (MWS 2, Berghof, Germany) [51]. Mineral content (N, P, K, Ca, Mg, S, Cu, Mn, Zn, B, active $Fe^{++}$, and total Fe) of spinach leaves and roots was determined using an inductively coupled plasma spectrophotometer (Optima 2100 DV; Perkin-Elmer, Shelton, CT, USA) [44,52,53].

### 2.8. Statistical Analysis

Data obtained from the measurements were evaluated statistically using analysis of variance (ANOVA). Means were separated by Duncan's multiple-range test. Regression analysis was performed to determine the most effective dose of Fe. Hyperbolic curves, graphs, and a heat map were prepared with GraphPad Prism 8.0 (GraphPad Software, Inc., La Jolla, CA, USA).

## 3. Results

In this study, the effects of different Fe doses, together with biostimulant applications, on the growth and quality of spinach were investigated. Both conventional and nano-sized Fe fertilizers were mixed with biostimulants, as given in Table 1. Figure 2 shows the changes in total Fe, active $Fe^{++}$, plant fresh weight, and chlorophyll reading value (SPAD) with the changes in Fe doses (0, 5, 10, and 20 mM) applied. Regression analysis was performed to determine the most effective dose of Fe, which was found to be 20 mM. For this reason, in the remainder of the study, the concentration of Fe in the treatment solutions was selected as 20 mM.

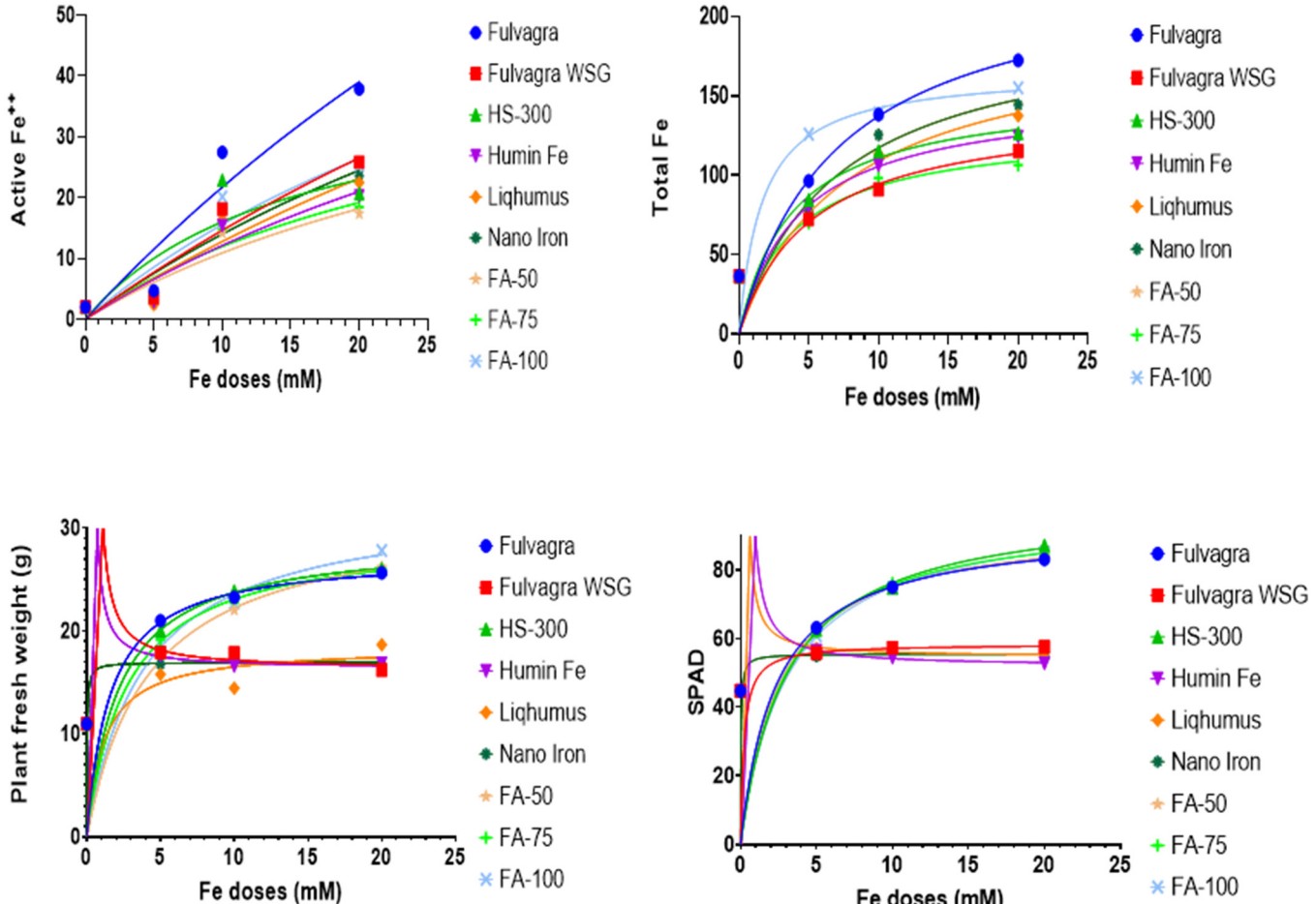

**Figure 2.** Regression analyses of different treatments to determine the most effective iron dose based on important agronomic properties: active Fe, total Fe, plant fresh weight, and SPAD value in spinach. Treatments Fulvagra, Fulvagra WSG, HS-300, Humin Fe, and Liqhumus contain $FeSO_4 \cdot 7H_2O$ as the iron source; treatments Nano Iron, FA-50, FA-75, and FA-100 contain nano ferrihydrite as the iron source.

### 3.1. Growth Properties and Chlorophyll and Carotenoid Contents

Figure 3 shows the changes in the plant fresh weight (g), plant dry weight (g), plant dry matter (%), root fresh weight (g), root dry weight (g), and dry matter (%) when different treatments composed of various biostimulants and Fe were applied to spinach. The fresh and dry weights of the plants and roots were significantly higher than those of the control ($p \leq 0.001$). Among the treatments containing $FeSO_4 \cdot 7H_2O$, treatments containing Fulvagra and HS-300; treatments containing Fulvagra and HS-300; among the treatments with nano Fe, all three (FA-50, FA-75, and FA-100) were determined as the most effective treatments in terms of plant fresh weight, plant dry weight, and dry matter content. The plant fresh weight, plant dry weight, and dry matter content of spinach increased by 60.99%, 56.25%, and 36.61%, respectively, for Fulvagra; 57.90%, 49.34%, and 21.76%, respectively, for HS-300; 57.90%, 48.21%, and 22.45%, respectively, for FA-50; 60.63%, 50.03%, and 22.45%, respectively, for FA-75; and 58.19%, 51.05%, and 24.22%, respectively, for FA-100.

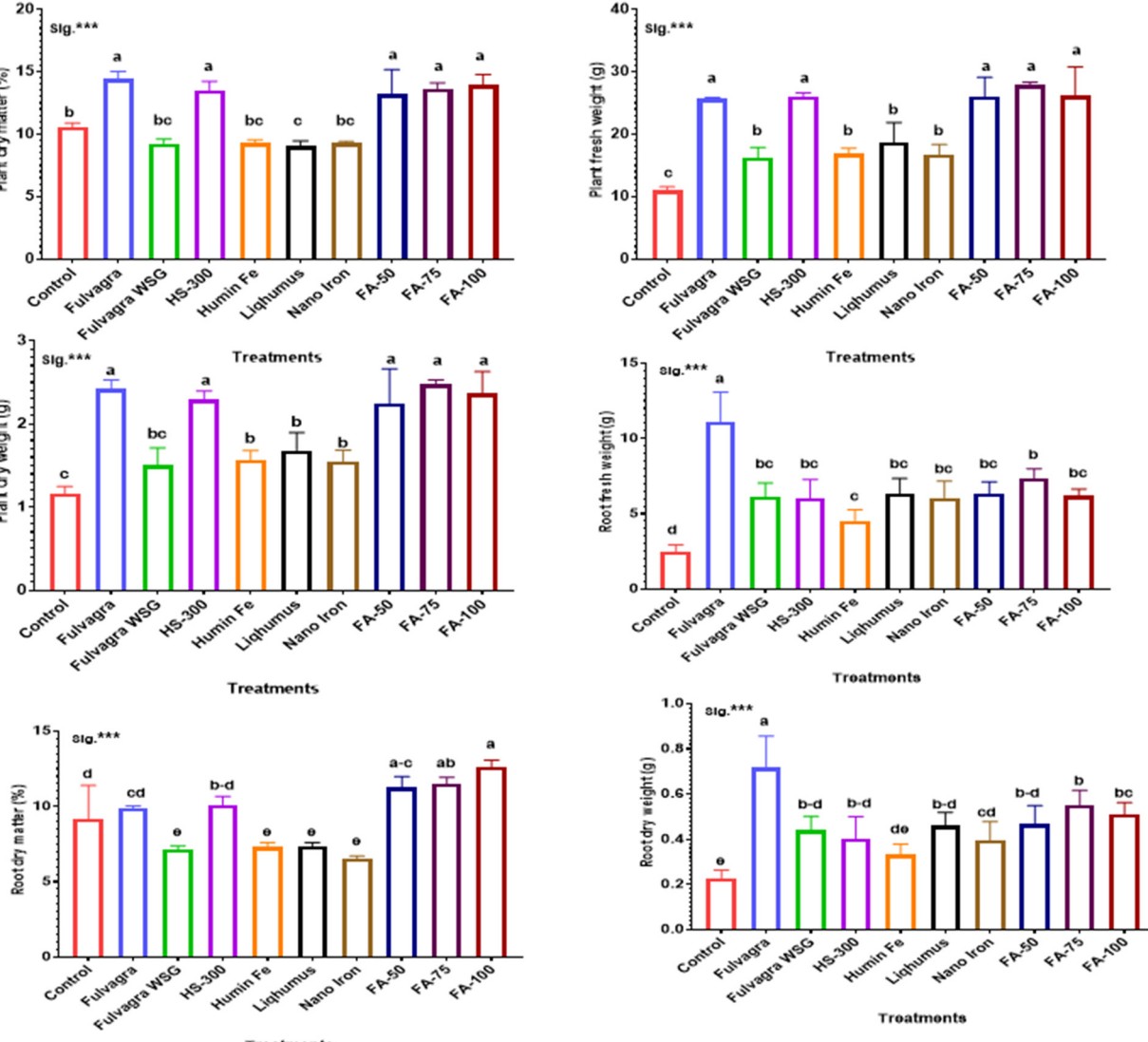

**Figure 3.** The effects of different treatments on plant fresh weight, plant dry weight, plant dry matter content, root fresh weight, root dry weight, and root dry matter content of spinach. Treatments Fulvagra, Fulvagra WSG, HS-300, Humin Fe, and Liqhumus contain $FeSO_4 \cdot 7H_2O$; treatments Nano Iron, FA-50, FA-75, and FA-100 contain nano ferrihydrite. The iron concentration in all treatments is 20 mM. The mean values with the same letter within the same column are not significantly different (***: $p > 0.001$).

When root fresh weight (3.08 g/plant) and root dry weight (0.27 g/plant) were compared with the others, the treatment containing Fulvagra was found to be more effective. On the other hand, root dry matter content was found to be higher for the applications containing fulvic acid and nano ferrihydrite, especially FA-100 (12.64%).

Compared with the control, the root diameter (mm), leaf number per plant, leaf area (cm²), and SPAD value were significantly affected ($p \leq 0.001$) by the treatments (Figure 4). The highest root diameter (7.20 mm) was obtained from the Fulvagra treatment, whereas the control had the lowest value (3.75 mm). The highest number of leaves (34.86 leaves per plant) was achieved with HS-300 treatment and the least number of leaves (16.45 leaves per plant) was found in the control group. When the leaf area per plant was considered, Fulvagra, HS-300, FA-50, FA-75, and FA-100 treatments gave the best results with 501.63, 497.48, 533.44, 496.31, and 470.54 cm², respectively, and increased by 69.88%, 69.63%, 71.68%, 69.56, and 67.89, respectively, compared with the control. The leaf color intensity

was found to increase with the treatments; the highest SPAD value (87.23) was found in the treatments containing HS-300 and the lowest (44.70) was obtained from the control.

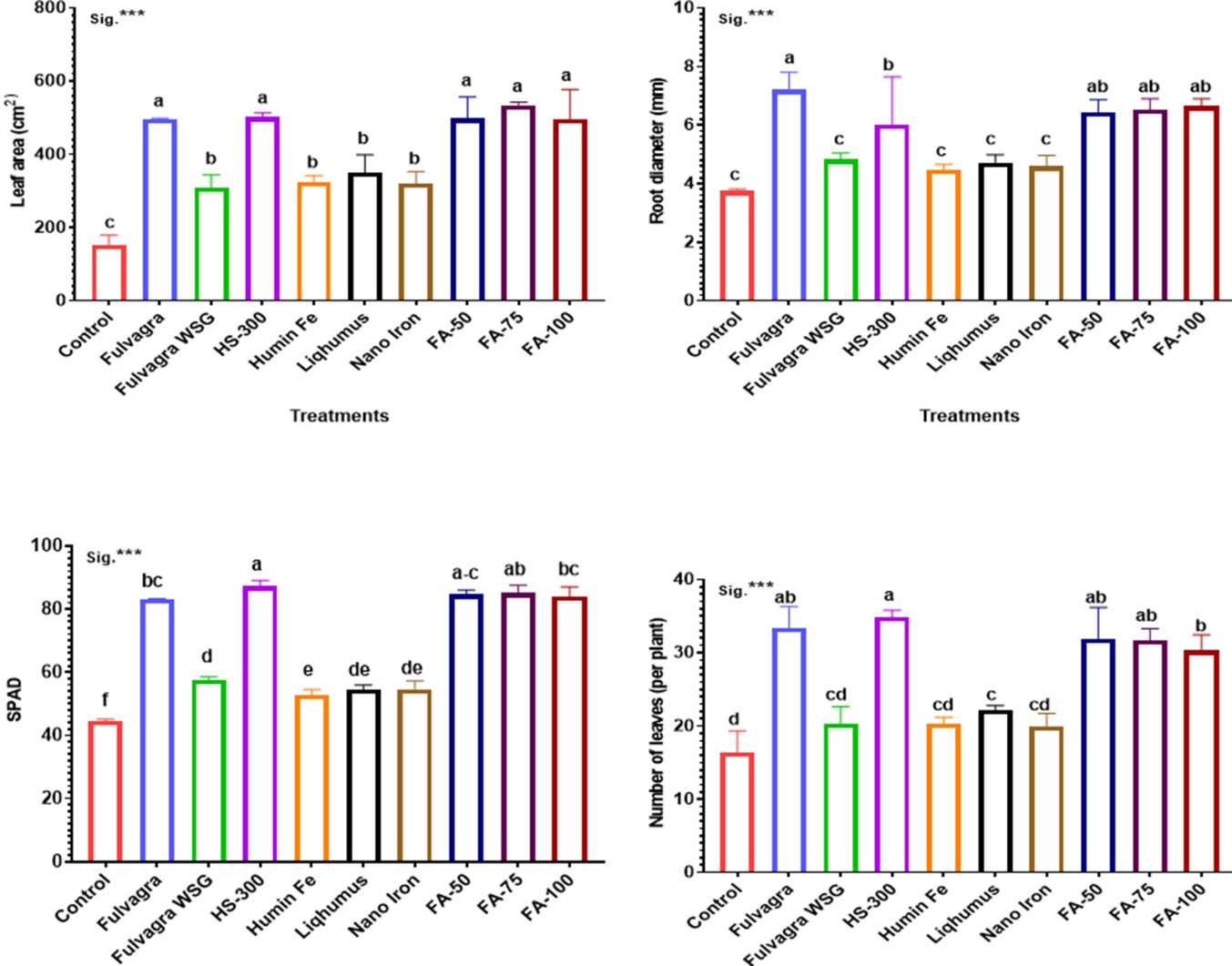

**Figure 4.** The effects of different treatments on root diameter, leaf number, leaf area, and SPAD value of spinach. Treatments Fulvagra, Fulvagra WSG, HS-300, Humin Fe, and Liqhumus contain $FeSO_4 \cdot 7H_2O$; treatments Nano Iron, FA-50, FA-75, and FA-100 contain nano ferrihydrite. The iron concentration in all treatments is 20 mM. The mean values with the same letter within the same column are not significantly different (***: $p > 0.001$).

Chlorophyll a, chlorophyll b, total chlorophyll, and total carotenoid contents of spinach were significantly affected ($p \leq 0.001$) by different treatments (Figure 5). Chlorophyll a, chlorophyll b, total chlorophyll, and total carotenoid contents values were 3.49, 1.67, 5.17, and 11.65 mg cm$^{-2}$, respectively, for FA-50; 3.73, 1.71, 5.44, and 11.66 mg cm$^{-2}$, respectively, for FA-75; 3.67, 1.77, 5.44, and 12.04 mg cm$^{-2}$, respectively, for FA-100. Chlorophyll a, chlorophyll b, total chlorophyll, and total carotenoid contents increased by 106.40%, 102.63%, 105.16%, and 93.25%, respectively, for FA-50; 120.12%, 107.40%, 115.90%, and 99.65%, respectively, for FA-75; and 116.76%, 114.67%, 116.08%, and 105.24%, respectively, for FA-100.

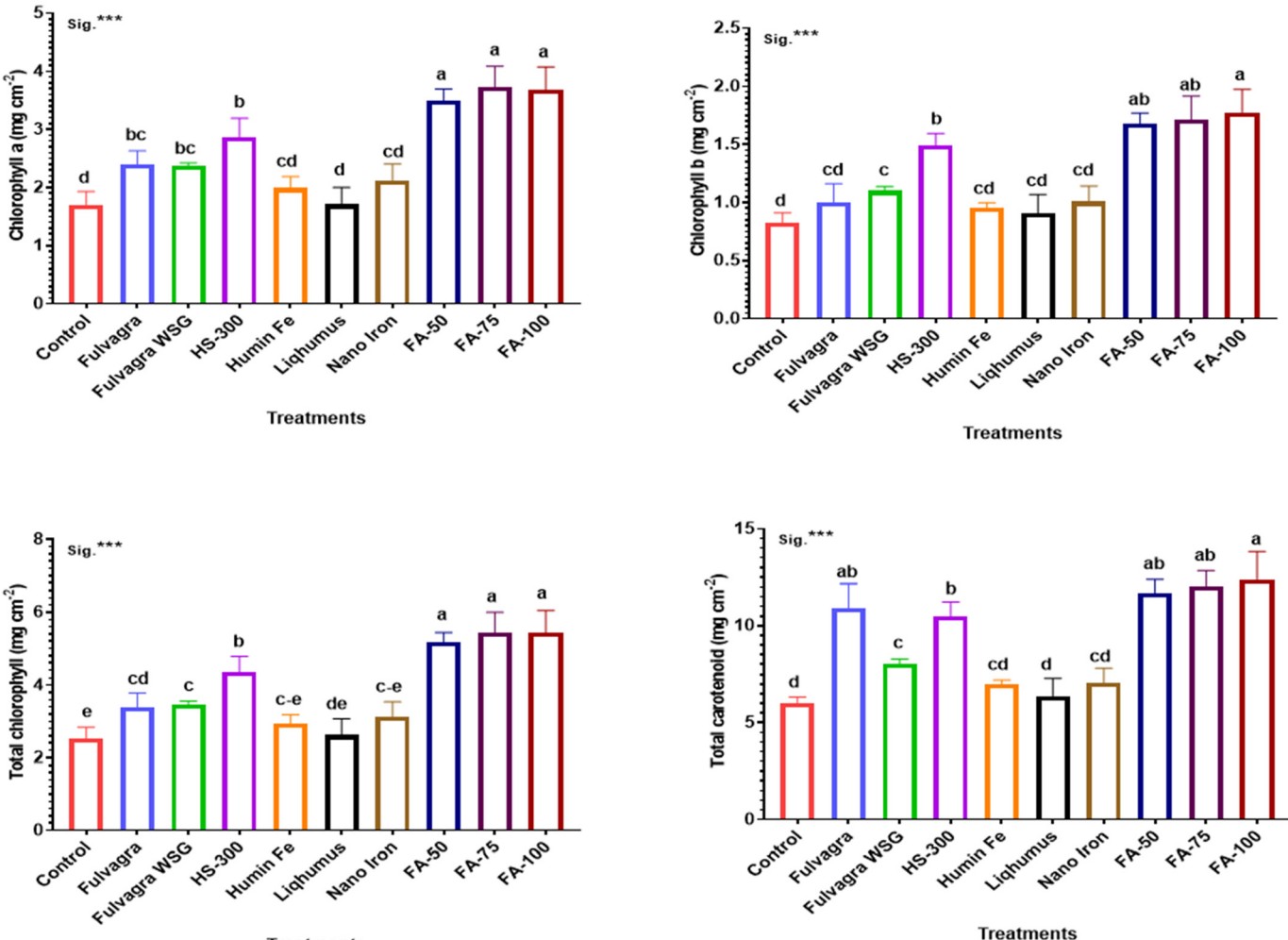

**Figure 5.** The effects of different treatments on chlorophyll a, chlorophyll b, total chlorophyll, and total carotenoid contents spinach. Treatments Fulvagra, Fulvagra WSG, HS-300, Humin Fe, and Liqhumus contain $FeSO_4 \cdot 7H_2O$; treatments Nano Iron, FA-50, FA-75, and FA-100 contain nano ferrihydrite. The iron concentration in all treatments is 20 mM. The mean values with the same letter within the same column are not significantly different (***: $p > 0.001$).

Figure 6 shows the heatmap analysis for percentage change (%) of the growth parameters, as well as the chlorophyll and carotenoid contents of spinach with different treatments compared with the control. On the basis of the heatmap analysis, the dry weight of plants (g) in FA-75 (113.62%), Fulvagra (108.77%), FA-100 (104.53%), HS-300 (98.01%), FA-50 (93.51%), and chelate (88.22%) treatments showed high increases compared with the control; plant dry matter content was also significantly increased in Fulvagra (36.57%), FA-100 (31.95%), FA-75 (28.86%), Fe chelate (28.24%), HS-300 (27.77%), and FA-50 (24.61%) treatments compared with the control. On the other hand, the application of Fulvagra biostimulant was the best treatment, exhibiting an increase of 344.95% and 220.00% in fresh weight and dry weight of roots, respectively, compared with the control.

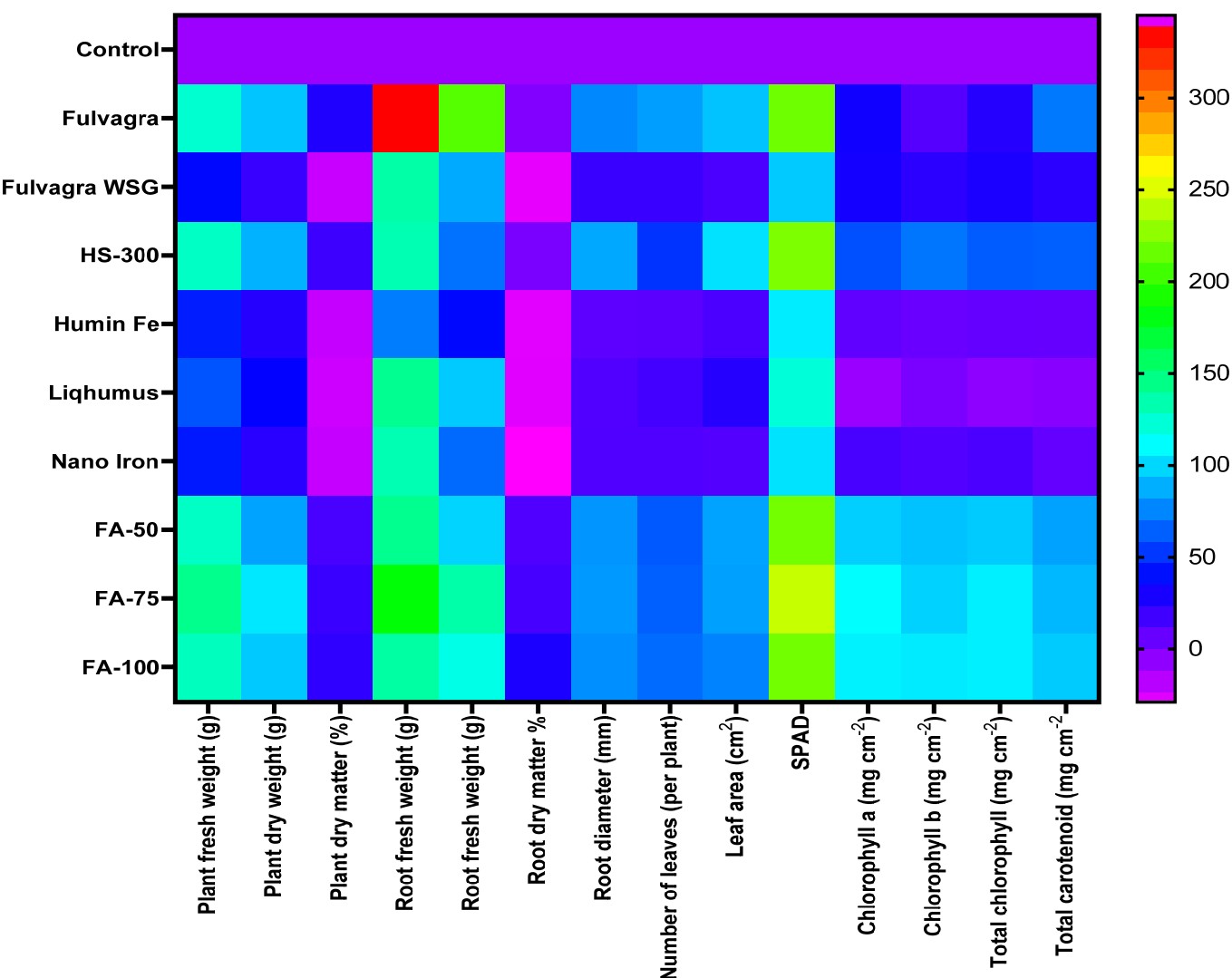

**Figure 6.** Heatmap analysis for percentage change (%) of the growth parameters and chlorophyll and carotenoid contents of spinach with different treatments compared with the control.

### 3.2. Leaf and Root Nutrient Content

Leaf and root N, P, K, Ca, Mg, S, Cu, Mn, Zn, B, active Fe, and total Fe contents were significantly affected ($p \leq 0.001$) by different treatments (Table 3). According to the data, the effects of treatments on leaf and root N concentrations varied.

Leaf N concentrations ranged from 2.66 to 4.18 mg kg$^{-1}$, whereas root N concentrations ranged from 0.97 to 1.53 mg kg$^{-1}$. Compared with the control, leaf N content increased by 20.26%, 27.51%, and 24.06%, with Fulvagra, HS-300, and Humin Fe treatments, respectively. Leaf P concentration varied between 0.25 and 0.42 mg kg$^{-1}$, and root phosphorus concentration varied between 0.12 and 0.21 mg kg$^{-1}$. The highest concentration of root and leaf P was detected in the treatment containing Humin Fe. The K content of roots and leaves varied between 1.59 and 3.69 mg kg$^{-1}$ and between 0.36 and 0.84 mg kg$^{-1}$, respectively. The P content for both fractions was highest in the HS 300 treatment and lowest in the FA-100 treatment. The highest leaf (2.19 mg kg$^{-1}$) and root (0.26 mg kg$^{-1}$) Ca contents were determined in FA-50 application. The lowest leaf and root Ca contents were obtained from Fulvagra WSG, Nano Iron, FA-100, and control treatments. The Mg content of both fractions was positively affected by all treatments compared with the control, whereas the most effective results were obtained from Humin Fe application. The highest and lowest Mg contents were 0.18 and 0.41 mg kg$^{-1}$, respectively, for leaves and 0.07 and 0.15 mg kg$^{-1}$, respectively, for roots. The S content varied between 0.18 and 0.38 mg kg$^{-1}$ in the leaves

and between 0.09 and 0.19 mg kg$^{-1}$ in the roots. The highest S content was achieved with HS-300 treatment in both roots and leaves, whereas the lowest S content was obtained from FA-75 treatment in leaves and from FA-50, FA-50, and FA-100 in roots fractions. The highest Cu concentration in the leaves was achieved when FA-50, FA-75, and FA-100 treatments were applied, whereas the lowest Cu content was obtained from Humin Fe and the control treatments. Leaf Mn concentrations ranged between 27.80 and 44.15 mg kg$^{-1}$, and root Mn concentrations were between 2.13 and 3.38 mg kg$^{-1}$. All treatments increased leaf and root Zn concentration significantly compared with the control, and the most effective treatment was FA-100 application (54.00 and 4.14 mg kg$^{-1}$, respectively). The highest B content was determined from the treatments containing HS-300 and Humin Fe for both leaves and roots. The leaf boron concentration varied between 5.26 and 18.14 mg kg$^{-1}$, and the root boron concentration ranged between 1.58 and 5.44 mg kg$^{-1}$. The highest B content in both leaves and roots was achieved with HS-300 and Humin Fe treatments, and the lowest B content was determined from FA-75 treatment.

**Table 3.** Effects of different treatments on leaf and root macro- and micro-nutrient content.

| Treatments | N (%) | P | K | Ca | Mg | S | Cu | Mn | Zn | B | Active Fe | Total Fe |
|---|---|---|---|---|---|---|---|---|---|---|---|---|
| | Leaf Nutrient Content (mg kg$^{-1}$) $^{\S}$ | | | | | | | | | | | |
| Control | 3.03 cd | 0.28 bc | 2.45 de | 1.08 e | 0.18 e | 0.30 bc | 12.54 d | 36.30 b–d | 13.52 f | 6.16 de | 1.97 f | 36.39 e |
| Fulvagra | 3.80 a | 0.34 e–g | 3.02 b | 1.62 d | 0.34 a–d | 0.29 bc | 21.26 bc | 32.60 c–f | 39.20 bc | 9.81 bc | 37.83 a | 163.09 a |
| Fulvagra WSG | 3.64 b | 0.25 g | 2.56 d | 1.12 e | 0.29 cd | 0.27 c | 26.17 b | 29.06 ef | 28.83 de | 10.06 bc | 25.81 b | 115.20 d |
| HS-300 | 4.18 a | 0.36 b | 3.69 a | 1.68 cd | 0.39 ab | 0.38 a | 17.25 cd | 41.00 ac | 34.41 cd | 18.14 a | 20.67 d | 126.63 c |
| Humin Fe | 3.99 a | 0.42 a | 3.15 b | 1.85 bc | 0.41 a | 0.31 b | 11.28 d | 44.15 a | 27.55 de | 15.78 a | 20.27 d | 124.17 c |
| Liqhumus | 3.06 cd | 0.31 c–e | 2.62 cd | 1.78 cd | 0.36 a–c | 0.23 d | 21.33 bc | 35.11 b–e | 31.53 d | 12.16 b | 22.44 c | 137.50 b |
| Nano Iron | 2.80 de | 0.30 d–f | 2.18 e | 1.24 e | 0.31 b–d | 0.21 de | 21.99 bc | 31.29 d–f | 45.88 b | 10.72 bc | 23.56 c | 144.32 b |
| FA-50 | 2.94 c–e | 0.34 bc | 2.97 b | 2.19 a | 0.30 cd | 0.20 d–f | 28.67 ab | 30.37 d–f | 24.17 e | 12.11 b | 17.36 e | 106.40 d |
| FA-75 | 2.66 e | 0.27 fg | 2.91 bc | 2.00 b | 0.26 d | 0.17 f | 35.26 a | 27.80 f | 42.81 b | 5.26 e | 18.43 e | 112.93 d |
| FA-100 | 2.76 de | 0.32 b–d | 1.59 f | 1.05 e | 0.29 cd | 0.18 ef | 35.59 a | 30.48 d–f | 54.00 a | 6.54 de | 25.32 b | 155.13 a |
| | Root Nutrient Content (mg kg$^{-1}$) $^{\S}$ | | | | | | | | | | | |
| Control | 1.04 de | 0.17 bc | 0.56 de | 0.13 e | 0.07 e | 0.15 bc | 2.86 d | 2.78 b–d | 1.04 f | 1.85 de | 0.16 g | 0.16 e *** |
| Fulvagra | 1.43 a | 0.13 e–g | 0.68 b | 0.19 d | 0.12 a–d | 0.14 bc | 4.84 bc | 2.50 c–f | 3.00 bc | 2.94 bc | 9.68 a | 27.60 a |
| Fulvagra WSG | 1.33 b | 0.12 g | 0.58 d | 0.14 e | 0.11 cd | 0.13 c | 5.96 b | 2.23 fe | 2.21 de | 3.02 bc | 7.74 b | 19.50 d |
| HS-300 | 1.53 a | 0.17 b | 0.84 a | 0.20 cd | 0.14 ab | 0.19 a | 3.92 cd | 3.14 ab | 2.64 cd | 5.44 a | 6.20 c–e | 21.43 c |
| Humin Fe | 1.46 a | 0.21 a | 0.72 b | 0.22 bc | 0.15 a | 0.15 b | 2.57 d | 3.38 a | 2.11 de | 4.73 a | 6.08 d–f | 21.01 c |
| Liqhumus | 1.12 cd | 0.15 c–e | 0.60 cd | 0.21 cd | 0.13 a–c | 0.11 d | 4.85 bc | 2.69 b–e | 2.42 d | 3.65 b | 6.73 cd | 23.27 b |
| Nano Iron | 1.03 de | 0.14 d–f | 0.50 e | 0.15 e | 0.11 b–d | 0.11 de | 5.00 bc | 2.40 d–f | 3.52 b | 3.22 bc | 7.07 bc | 24.43 b |
| FA-50 | 1.08 c–e | 0.17 bc | 0.68 b | 0.26 a | 0.11 cd | 0.09 d–f | 6.52 ab | 2.33 d–f | 1.85 de | 3.63 b | 5.21 f | 18.01 d |
| FA-75 | 0.97 e | 0.13 fg | 0.67 bc | 0.24 b | 0.09 d | 0.09 d–f | 8.02 a | 2.13 ef | 3.28 b | 1.58 e | 5.53 ef | 19.11 d |
| FA-100 | 1.01 de | 0.16 b–d | 0.36 f | 0.13 e | 0.10 cd | 0.09 d–f | 8.10 a | 2.34 d–f | 4.14 a | 1.96 de | 8.93 a | 26.26 a |

$^{\S}$ The mean values with the same letter within the same column are not significantly different (***: $p > 0.001$).

Table 3 shows the active and total Fe contents of spinach leaves and roots. All treatments significantly affected the active and total Fe contents of leaves and roots ($p < 0.001$). Fulvagra-treated spinach had the highest active and total Fe values both in leaves (37.83 mg kg$^{-1}$ and 163.09 mg kg$^{-1}$, respectively) and roots (9.68 mg kg$^{-1}$ and 27.60 mg kg$^{-1}$, respectively). Active and total Fe contents increased by 18.20-fold and 4.20-fold for leaves and by 60-fold and 6.04-fold for roots, respectively.

## 4. Discussion

Iron is an important element for the plant growth and the level of the requirement by the plant is often an issue in numerous crops, particularly in calcareous soils. Iron is a prosthetic constituent of ferredoxin, a critical protein for chlorophyll synthesis and photosynthesis in plants [5]. Thus, Fe deficiency leads to interveinal chlorosis on young leaves, which is the yellowing/whitening of the leaves between the veins due to abnormal chlorophyll synthesis [4]. The reduction in leaf chlorophyll content caused by Fe deficiency

is often characterized by the reduced leaf area and total plant dry weight, resulting in lower yields and crop quality [54–57].

Our results showed that foliar treatments containing fulvic acid together with nano ferrihydrite (FA-50, FA-75, and FA-100) resulted in higher chlorophyll a, chlorophyll b, and total chlorophyll contents (Figure 5). The leaf area was also significantly higher for these treatments (Figure 4). High leaf area was also obtained with Fulvagra and HS-300 treatments. For these five treatments (FA-50, FA-75, FA-100, Fulvagra, and HS-300) the SPAD value, which indicates the green color intensity of the leaf, was also higher. The highest total Fe values were found in the leaves to which Fulvagra and FA-100 were applied (Table 3). More importantly, the highest active Fe (37.83 mg/kg) in the leaves was obtained from Fulvagra-treated spinach, followed by Fulvagra WSG (25.81 mg kg$^{-1}$) and FA-100 (25.32 mg kg$^{-1}$). The amount of active Fe is critical because even the plants having total Fe content within the adequacy limits suffer from Fe deficiency if their active Fe content is low.

Total carotenoid content, which is important for the plant's immune system, was found to be the highest for the same five treatments (FA-50, FA-75, FA-100, Fulvagra, and HS-300) (Figure 5). The increase in the chlorophyll content, leaf number, leaf area, SPAD value, and active Fe content of leaves correlated well with the growth parameters and the yield. Plant fresh weight, plant dry weight, and plant dry matter for the same five treatments were significantly higher than the rest (Figure 3). Fulvagra treatment was found to be the most efficient treatment in increasing the root fresh weight and root dry weight (Figure 3). These results are promising because, as has been reported earlier, growth of roots and above-ground fractions decrease significantly under Fe-deficient conditions [58]. Compared to the control, the micro- and macroelement contents of leaves and roots were found to increase significantly with the treatments. When all parameters in the study were considered, Fulvagra treatment—which contained 17% fulvic acid and microorganisms in its content together with 20 mM FeSO$_4$·7H$_2$O—and fulvic acid-based nano ferrihydrite—containing treatments (FA 50, FA 75, and, especially, FA100 treatment which contained 67.5–69.5% fulvic acid)—were found to be the most effective treatments. This indicates that fulvic acid containing biostimulants are more effective in foliar applications against Fe deficiency.

Earlier studies have already reported that applying Fe sources and Fe chelates to the soil or the leaves decreases the adverse effects of Fe deficiency [7,13,15,16]. Moreover, many studies also showed that applying biostimulants to the soil or to the leaves is an environmentally friendly way of managing Fe deficiency problems [27–29,34,38,59]. Our results showed that: (1) foliar application of biostimulants together with Fe sources improve the nutrient uptake, chlorophyll contents, growth characteristics, and yield and (2) not all humic substances (HS, humic acid, and fulvic acid) have the same effect.

HS are composed of molecules (humic acid, fulvic acid, humin, etc.) at different molecular weights and solubilities. HS improve the plant growth via various mechanisms. Although the foliar and soil applications of HS have been shown to increase the yields, quality, and productivity of many crops and fruit trees [32,35,36,39], a more common way of utilizing HS is the soil applications. It has been shown in many studies that when applied to the soil, HS (due their unique composition) promote the uptake, assimilation, and distribution of macro- and microelements, including Fe, in roots and shoots which stimulate the plant growth [32,60–65]. When they are taken up by plants, some low-molecular-weight compounds in HS increase the cell membrane permeability, which facilitates the nutrient uptake in a manner similar to that of hormones [60,62]. Moreover, HS stimulate H+-ATPase activity and support secondary ion transporters, which also increases the nutrient uptake [63]. However, increasing the nutrient uptake is not the only reason that HS are effective under Fe-deficient environments. In calcareous soils, the solubility of Fe is very limited. HS are redox-reactive and can reduce Fe$^{3+}$ into soluble forms [66,67]. Moreover, owing to the high number of carboxyl and phenolic hydroxyl groups in their structure, HS have high cation exchange capacities (CEC) and form strong bonds with metals [68]. For this reason, HS form Fe–HS complexes that can act as natural Fe chelates, where they

transport to the roots to be used by the plants even in calcareous soils with limited Fe availability [69–71]. Although complexes with low-molecular-weight HS are as soluble and mobile as Fe chelates and can be readily used by roots, complexes with high-molecular-weight HS are mainly insoluble and form an Fe reserve in soil that can be taken up by root interception or by ligand exchange [27].

In addition to the soil applications, foliar spraying of fulvic acid has been reported to facilitate the transport of nutrients from roots to shoots—particularly the elements involved in photosynthesis, such as Fe, Zn, and Mn—and to increase ROS-scavenging capacity to cope with stress and facilitate the photosynthetic activity, which in turn promotes the plant growth [72]. Fulvic acid might be considered more efficient for foliar applications because fulvic acid has a relatively low molecular weight compared with other humic substances. Our results are in accordance with the earlier findings that the most efficient two treatments in our study, namely, Fulvagra and FA-100, contained higher amounts of fulvic acid than humic acid in their content.

Foliar applications were also reported as a more efficient way for the application of nanoparticles [73,74]. Delivery of Fe nanoparticles via leaf spray as compared with soil amendments showed that foliar applications have increased nutrient uptake and bioavailability [2,19–22]. HS were shown to improve Fe uptake via foliar applications [30,31,33]; however, the number of studies on foliar applications of HS together with nano Fe particles is scarce [40]. Our results showed that applications composed of ferrihydrite nanoparticles and fulvic acid (FA-50, FA-75 and FA-100) had better results in terms of plant growth parameters compared with nano iron treatments which contains high amount of humic acid (73.5%) with ferrihydrite nanoparticles. Total Fe and active Fe amounts were also significantly higher for FA-100 compared to the Nano Iron treatment. This finding once again suggests that fulvic acid foliar treatments are superior in performance to humic-based foliar applications. On the other hand, using Fe nanoparticles instead of conventional Fe fertilizers did not have an additional positive affect.

In addition to the synergistic effect of Fe and fulvic acid in Fulvagra, the plant growth-promoting rhizobacteria (PGPR) in its content might also have a beneficial effect on the yield parameters and active and total Fe content of the plant. Many studies have shown that foliar applications of PGPR increases the yield, growth, and nutrient element content, including Fe, of crops and fruit trees [75–79]. Once PGPR are applied to the leaves, the phytohormones or active metabolites produced by the bacteria have been reported to cause changes in the hormonal balance of the plant, which positively affect the plant growth. Moreover, the synergistic relationship between PGPR and HS in Fulvagra might also have contributed to the Fe acquisition mechanism and plant growth. On the other hand, the underlying reason of not obtaining similar stimulating effects from Fulvagra WSG treatment despite its high fulvic acid content might be explained by the loss of the low-molecular-weight components and the changes in its structure during the drying step in its manufacturing process.

## 5. Conclusions

The availability of Fe is reduced in calcareous soils due to high pH or high concentration of phosphorus. Iron deficiency causes disorders in photosynthetic activity and decreases uptake of other nutrient elements. Although providing the required nutrients to the plants is essential, inappropriate soil fertility management has a large negative impact on the environment, such as deterioration of soil health, environmental pollution, and the emergence of pathogen and pest populations. Supplementing soil with non-biodegradable synthetic Fe chelates is both uneconomical and unfriendly to the environment due to leaching. Foliar applications are a greener alternative to soil amendments to cope with Fe deficiency. In this study, we sprayed HS together with $FeSO_4 \cdot 7H_2O$ or with nano ferrihydrite on the leaves to increase the Fe content and yield parameters of spinach. When all treatments were compared, Fulvagra treatment, which contained fulvic acid, plant growth-promoting rhizobacteria (PGPR), and $FeSO_4 \cdot 7H_2O$ in its content, was found to be the most

efficient treatment in terms of the growth parameters and active and total Fe content of the plant. Fulvagra foliar application was followed by FA-100 treatment, which was composed of fulvic acid and nano ferrihydrite. Based on these findings, low-molecular-weight fulvic acid was more efficient in foliar application than other HS. The study does not provide evidence for the induction of plant growth in various environmental conditions. Thus, further studies are required to determine the effect of treatments and the efficiency of these materials under natural field conditions and other plant species.

**Author Contributions:** M.T., M.E. and E.Y. designed the experiments; M.T., M.E., R.K. and E.Y. conducted the experiments; M.T., M.E., R.K., E.Y., S.A., A.K., A.M.Z. and I.V.P. analyzed the results. All authors have read and agreed to the published version of the manuscript.

**Funding:** This research received no external funding.

**Institutional Review Board Statement:** Not applicable.

**Informed Consent Statement:** Not applicable.

**Data Availability Statement:** Not applicable.

**Acknowledgments:** Authors would like to acknowledge the support under the state task 121021000105-7 and the support of scientific-educational school of the Lomonosov MSU "Future of the planet and global change of the environment". Also, we are very grateful to Humintech GmbH for generous support.

**Conflicts of Interest:** The authors declare no conflict of interest.

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
