# Peer review of "Foliar Applications of Humic Substances Together with Fe/Nano Fe to Increase the Iron Content and Growth Parameters of Spinach (Spinacia oleracea L.)"

_agronomy, doi:10.3390/agronomy12092044_

Round 1

Reviewer 1 Report

The manuscript titled "Foliar Applications of Humic Substances Together with Fe/nano Fe to Increase the Iron Content and Growth Parameters of Spinach (Spinacia oleracea L.)" presents an interesting research topic. The title, abstract and keywords generally reflect well on the manuscript as a whole. The subsections material and methods once results need to be improved. The discussion reflects well the content of the research and places the presented results among them. The conclusions are very general.

Major comments:

Why does chlorosis appear in the keywords? The text of the paper does not refer to it. 

In addition, in the materials and methods, there are in part the results of the study. Please separate these data, and move the results on preliminary analyses to the beginning of the results section. 

Line 144: PDF-2 database? 

Please detail the conclusions of the experiment. Indicate further possibilities for research and use of the results. 

Line 258 says that regression analysis was used, and this is not in the methodology. 

There is no description of the data contained in the heat map and no indication of the purpose why it was made. 

Table 3 is missing. 

Other comments:

When stating ions, please use superscript. 

When giving sum formulas of chemical compounds, please use subscript.

Please give numerical values to significant places, usually 3 significant places are accepted. 

Latin names, please write in italics. 

Please give the content on figures in the same font as the whole content of the manuscript and get rid of borders.

In Table 1, pH values are given differently (positions 2 and 9), but this is a minor technical note. 

In line 133, please use the correct citation. 

Please correct the references according to the requirements of the journals.

Overall, I consider the content of the manuscript and the study design to be good, I recommend rethinking the presentation of the content and separating the methodology from the results, and improving the conclusions. In addition, I recommend quietly reviewing the manuscript to get rid of shortcomings. 

Author Response

Dear Editor,

On behalf of all of the co-authors, I would like to thank the editor and the reviewers for careful reading, and constructive suggestions for our manuscript. According to comments from editor and reviewers, we comprehensively revised our manuscript. We are looking forward to your feedback and further comments/suggestions are welcome.

The reviewers' comments are reproduced in black; our responses are detailed below in red:

RESPONSE TO REVIEWERS  First of all, I would like to extend our gratitude to the anonymous reviewers for forwarding very constructive criticism of the MS and giving us an opportunity to improve the standard of the MS. We are glad to state that all suggestions forwarded by the reviewers have been incorporated in their entirety and none has been left unaddressed.

Reviewer #1: The manuscript titled "Foliar Applications of Humic Substances Together with Fe/nano Fe to Increase the Iron Content and Growth Parameters of Spinach (Spinacia oleracea L.)" presents an interesting research topic. The title, abstract and keywords generally reflect well on the manuscript as a whole. The subsections material and methods once results need to be improved. The discussion reflects well the content of the research and places the presented results among them. The conclusions are very general.
Thank you for the suggestion and agreeing to reviewer’s comments. The work is of good novelty. Thank you again for the appreciation of the work.

Comments
1- Why does chlorosis appear in the keywords? The text of the paper does not refer to it.

Agreeing to reviewer’s suggestion, we have deleted the word “chlorosis”

2- In addition, in the materials and methods, there are in part the results of the study. Please separate these data, and move the results on preliminary analyses to the beginning of the results section.

Agreeing to reviewer’s suggestion, we have moved some data to the results section.

3- Line 144: PDF-2 database?

PDF-2 features a FREE stand-alone option using ICDD’s integrated data-mining software, along with ICDD’s search-indexing software, SIeve.  Designed for inorganic materials analyses, PDF-2 also includes common organic materials from ICDD to facilitate rapid materials identification.

4- Please detail the conclusions of the experiment. Indicate further possibilities for research and use of the results.

Thank you very much for highlighting very important point. Agreeing to reviewer’s suggestion, we have added some statements to the conclusion section.

4- Line 258 says that regression analysis was used, and this is not in the methodology.

Thank you very much for highlighting very important point. Agreeing to reviewer’s suggestion, we have added the regression analysis in the methodology.

5- There is no description of the data contained in the heat map and no indication of the purpose why it was made.

Thank you very much for highlighting very important point. Agreeing to reviewer’s suggestion, we have added the reference as requested by Reviewer.  

6- Table 3 is missing.

We apologize for this inconvenience please see that we have added to Table 3.

Other comments
When stating ions, please use superscript.

Done

When giving sum formulas of chemical compounds, please use subscript.

Done

Please give numerical values to significant places, usually 3 significant places are accepted.

Done

Latin names, please write in italics.

Done

Please give the content on figures in the same font as the whole content of the manuscript and get rid of borders.

Done

In Table 1, pH values are given differently (positions 2 and 9), but this is a minor technical note.

Done

In line 133, please use the correct citation.

Done

Please correct the references according to the requirements of the journals.

 Done

Overall, I consider the content of the manuscript and the study design to be good, I recommend rethinking the presentation of the content and separating the methodology from the results, and improving the conclusions. In addition, I recommend quietly reviewing the manuscript to get rid of shortcomings.

Agreeing to reviewer’s suggestion, we have made the corrections.

The manuscript has been revised according to the Reviewers’ comments and Journal style,

So, we are looking forward to hearing good news from you.

Reviewer 2 Report

The work entitled “Foliar Applications of Humic Substances Together with Fe/nano Fe to Increase the Iron Content and Growth Parameters of Spinach (Spinacia oleracea L.)” fits with the aim of the journal Agronomy MDPI. In this work, the authors investigated the combined effects of Fe/nano-Fe application and humic/fulvic acid-based biostimulant foliar applicationts on spinach plants growth parameters and Fe content in Fe-deficient soil. The manuscript is interesting since it deals with a very important problem, suggesting a green solution. The entire manuscript has formatting issues, please follow the authors guidelines on the Agronomy MDPI site. In the whole manuscript, superscripts and subscripts of formulas and units of measurement are missing, please correct them.

Abstract

The description of the treatments need to be rewritten, it is not clear.

Introduction

The introduction section is well written, organized and properly referenced. I have only a few suggestions:

Line 93 “Spinacia oleracea” must be italicized.

Authors need to include more information about humic substances (composition and effect) and spinach (origin, use, composition, etc.).

Materials and methods

Lines 100,101 “Spinacia oleracea” must be italicized.

Paragraph 2.1 is not clear. I don’t understand the treatments used, do you try all Fe doses in a separate experiment and then use the best in another one or do you determine the best dosage in a single experiment? If so, after how many applications?

Moreover, it is not clear if the Fe/nano Fe treatments were combined with all humic substances or only with a part of them. I am confused because it seems that you use different kinds of fulvic and humic acids. Please clarify this part, describing in detail everything.

216 When you talk about 6 replicates, do you mean that 6 leaves were used or that you made the measurement 6 times the same leaf?

226 “…were calculated as follows using the method of Arnon [42] and Wellburn 225 [43]:” I think that something is missing here, please provide the missing part.

247 the formula is missing.

Paragraph 2.8 is not satisfactory, the authors need to be more specific. Moreover, they performed a heatmap analysis, but I didn’t find how it was done.

Results

Results need to be rewritten in an easier way, they are full of numbers and percentages and are not clear.

The heatmap analysis was not described.

Discussion

The discussion section is not sufficient, authors often provide only a description of the results. Thus, this section needs to be deeply argumented.

References

503 reference number 1 is wrong, please adjust it

All References need to be formatted as recommended by the author's guidelines.

Final remarks

The manuscript has some flaws that need to be addressed, thus I recommend to accept the paper only after the major revisions I suggest.

Reviewer 3 Report

The paper has significant interest to stablish the feasibility of Foliar Applications of Humic Substances Together with Fe/nano Fe to Increase the Iron Content and Growth Parameters of Spinach. However, some important issues should be addressed:
1. Authors will check the instructions for authors.
2. Units have been corrected.

Reviewer 4 Report

Please, follow my comments and suggestions, there are much more references as you cited. Table 3. is missing!!!

Author Response

(The authors gave the same response as above.)
